# D-Ribose-Induced Glycation and Its Attenuation by the Aqueous Extract of *Nigella sativa* Seeds

**DOI:** 10.3390/medicina58121816

**Published:** 2022-12-09

**Authors:** Prairna Balyan, Mohammad Shamsul Ola, Abdullah S. Alhomida, Ahmad Ali

**Affiliations:** 1Department of Life Sciences, University of Mumbai, Mumbai 400098, India; 2Department of Biochemistry, College of Science, King Saud University, Riyadh 11451, Saudi Arabia

**Keywords:** advanced glycation end products (AGEs), diabetes, D-ribose, medicinal plant, *Nigella sativa*, protein aggregation

## Abstract

*Background and Objectives:* Glycation and oxidative stress are the major contributing factors responsible for diabetes and its secondary complications. Aminoguanidine, a hydrazine derivative, is the only approved drug that reduces glycation with its known side effects. As a result, research into medicinal plants with antioxidant and antiglycation properties is beneficial in treating diabetes and its consequences. This investigation aimed to examine the efficacy of the aqueous extract of *Nigella sativa* seeds against the D-ribose-induced glycation system. *Materials and Methods:* The suppression of α-amylase and α-glucosidase enzymes were used to assess the antidiabetic capacity. UV–Visible, fluorescence, and FTIR spectroscopy were used to characterize the *Nigella sativa* seed extract and its efficacy in preventing glycation. The inhibition of albumin glycation, fluorescent advanced glycation end products (AGEs) formation, thiol oxidation, and amyloid formation were used to evaluate the extracts’ antiglycation activity. In addition, the extent of glycoxidative DNA damage was analyzed using agarose gel electrophoresis. *Results:* The IC_50_ for the extract in the α-amylase and α-glucosidase enzyme inhibition assays were approximately 1.39 ± 0.016 and 1.01 ± 0.022 mg/mL, respectively. Throughout the investigation, it was found that the aqueous extract of *Nigella sativa* seeds (NSAE) inhibited the level of ketoamine, exerted a considerable drop in fluorescence intensity, and reduced carbonyl production and thiol modification when added to the D-ribose-induced glycation system. In addition, a reduction in the BSA-cross amyloid formation was seen in the Congo red, thioflavin T assay, and electrophoretic techniques. NSAE also exhibited a strong capability for DNA damage protection. *Conclusion:* It can be concluded that *Nigella sativa* could be used as a natural antidiabetic, antiglycation treatment and a cost-effective and environmentally friendly source of powerful bioactive chemicals.

## 1. Introduction

Chronic hyperglycemia promotes non-enzymatic protein glycation, which causes a series of cascade reactions involving the reducing sugars (carbonyl groups) and proteins (amino groups), resulting in the production of a reversible molecule known as Schiff’s base [1]. It is well known that Schiff’s base goes through numerous condensation and re-arrangement steps to form relatively stable Amadori products, which then undergo additional oxidative alterations to produce advanced glycation end products (AGEs), such as crosslinking fluorescent AGEs like pentosidine and non-fluorescent AGEs like *N*-(carboxymethyl) lysine (CML) [2]. Numerous studies have implicated protein glycation in the etiology of atherosclerosis, diabetes, end-stage renal disease, and neurodegenerative disease [3].

In euglycemic adults, however, no significant relationships between postprandial indicators of blood glucose control and glycated albumin have been identified [4], which suggests that D-ribose may cause protein glycation in addition to D-glucose. Therefore, figuring out whether D-ribose plays a role in the creation of glycated serum protein will aid our understanding of diabetes mellitus. D-ribose, a pentose monosaccharide, is found in all living cells. Although Ribose has a stronger reactivity with proteins than other sugars, such as glucose, it produces AGEs more quickly, which are researched extensively in glycation research [5]. In addition, glycation with D-ribose causes BSA to misfold quickly and form globular amyloid-like aggregations, which are critical in brain cell cytotoxicity [6]. As a result, it is crucial to determine the extent of D-ribose involved in glycation.

Aminoguanidine, a well-known standard antiglycation drug, has received the greatest attention since it suppresses the development of AGEs both in vitro and in vivo. However, current research suggests that aminoguanidine may cause harm when used to treat diabetic nephropathy [7]. As a result, substantial effort has been expended in the quest for natural-source medicinal molecules that successfully prevent AGE development.

Herbal medicines with antiglycation and antioxidant activity have been crucial for preventing and alleviating AGE-mediated diabetes problems [8]. According to early findings, *Nigella sativa* (*N. sativa*) seeds and their bioactive compounds suppress protein glycation in bovine serum albumin [9]. *N. sativa* is native to the eastern Mediterranean, the Indian subcontinent, northern Africa, and southwest Asia [10]. In addition to being well-known for its culinary uses, it has a long history of use in traditional medicine. In traditional medicine, *N. sativa* has been recommended for various illnesses and disorders, such as hypertension, anorexia, amenorrhea, paralysis, dermatitis, and bronchitis [11]. These traditional applications of *N. sativa* are primarily attributed to its various medicinal benefits, including antidiabetic, anti-inflammatory, antihypertensive, antioxidant, antimicrobial, immunomodulatory, cardioprotective, anticancer, neuroprotective, nephroprotective, hepatoprotective, and gastroprotective properties [12,13]. Therefore, this plant has been extensively investigated over the past few decades. The black cumin plant is considered one of the most promising options for preventing and treating diabetes because of its potent ability to lower blood sugar by raising insulin levels [14]. This is one of the first molecular studies to show that *N. sativa* aqueous extract (NSAE) inhibits D-ribose-mediated glycation of BSA and downstream processes such as protein aggregation and glycoxidation. According to our previous research, NSAE has high phytochemical content, antioxidant activities, and anti-inflammatory activity compared to other extracts such as methanolic, ethanolic, and hexane extracts [15]. Therefore, it is interesting to investigate the antiglycation potential of *Nigella sativa* aqueous extract (NSAE) against the D-ribose-mediated system.

## 2. Materials and Methods

### 2.1. Plant Material and Preparation of Plant Extract

*N. sativa* seeds were provided by Dr. Aqueela Sattar, Royal College, Mumbai, Maharashtra, with taxonomic recognition by the Institute of Herbal Science, Plant Anatomy Research Centre, Chennai, India (Certificate no. PARC/2019/3911). First, the seeds were rinsed three times with distilled water and dried in the laboratory at 37 °C on blotting paper to remove the moisture content. Then, the seeds were pulverized and extracted using a Soxhlet apparatus, taking water as a solvent. Then, the extract was concentrated by evaporating the extraction solvent using a thermostat. Finally, the extract obtained was weighed, and the percentage yield was estimated as follows:% yield = (Dry weight of extract ÷ Dry weight of plant material) × 100.

### 2.2. Phytochemical Analysis

#### 2.2.1. Determination of Total Phenolic Content (TPC)

The total phenolic content of NSAE was estimated using the Folin–Ciocalteu (FC) method, as described earlier [16]. A calibration curve was made using gallic acid (5–500 µg/mL), and the results were expressed as milligrams of gallic acid equivalent per gram of dry weight of extract (mg GAE/g DW).

#### 2.2.2. Determination of Total Flavonoid Content (TFC)

The test sample’s total flavonoid content was determined using the aluminum chloride method based on the previous method described by Aryal and colleagues [17]. Quercetin (0–50 µg/mL) was used to make the calibration curve. The total flavonoid content was expressed as milligrams of quercetin equivalent per gram of dry weight of extract (mg QE/g DW). 

### 2.3. Enzyme Inhibition Assays

#### 2.3.1. Inhibition of α-Amylase 

The modified approach reported by Sathiavelu and his co-workers [18] investigated the inhibition of α-amylase by NSAE (1000 µg/mL). Acarbose (0–1000 µg/mL) was used as standard. The alpha-amylase inhibitory activity was calculated as follows:(AC − AE/AC) × 100 
where AC is the absorbance of the control (without NSAE) and AE is the absorbance of the test sample (with NSAE).

#### 2.3.2. Inhibition of α-Glucosidase

The effect of NSAE (1000 µg/mL) on α-glucosidase activity was performed by the previous approach [19]. Acarbose (0–1000 µg/mL) was used as standard. The α-glucosidase inhibitory activity was calculated as follows: (AC − AE/AC) × 100 
where AC is the absorbance of the control (without NSAE) and AE is the absorbance of the test sample (with NSAE).

### 2.4. In Vitro Glycation of BSA Induced by D-Ribose

The glycated BSA formation induced by D-ribose was conducted using an earlier approach [9]. BSA (10 mg/mL) was incubated with 10 mg/mL D-ribose in phosphate buffer (0.1 M, pH 7.4) containing 0.02% sodium azide at 37 °C for four weeks. Aminoguanidine (AG) was used as a positive control at a concentration of 1 mM.

#### 2.4.1. Investigation of Hyperchromicity Utilizing UV–Vis Spectroscopy

Using a wavelength of 200–600 nm, the spectral analysis of all the samples with and without NSAE (100 µg/mL) was investigated [20]. The test sample’s glycation inhibitory activity was assessed using the equation below; findings were reported as a percentage of hyperchromicity at 280 nm.

% increase/decrease in hyperchromicity = Absorbance of glycated sample − Absorbance of native orNSAE-treated sample/Absorbance of glycated sample × 100.

#### 2.4.2. Fourier Transform Infrared Spectroscopy

Fourier transform infrared spectroscopy (FTIR) experiments were performed using an Pvt. FTIR RFPC 5301 (PerkinElmer, Waltham, MA, USA) spectrometer (4000 to 400 cm^−1^). The peaks obtained were examined using a standard IR spectra table.

#### 2.4.3. Measurement of Browning

The extent of browning was measured in all the samples using Shimadzu UV-1800 Spectrophotometer (Shimadzu, Japan) at 420 nm [21]. 

#### 2.4.4. Determination of Ketoamine

Throughout incubation, the formation of ketoamine or early glycation end products, i.e., Amadori product, was measured using nitroblue tetrazolium (NBT) assay [22]. The formation of ketoamine content was calculated using 1-deoxy-1-morpholino-fructose (1-DMF) as the standard.

#### 2.4.5. Determination of Protein Carbonyl Content

After the incubation period, the protein carbonyl content, a marker for oxidative protein damage, was assessed using an earlier approach reported by Levine and colleagues (1990) [23]. Based on the extinction coefficient for DNPH (ϵ = 22,000 M^−1^cm^−1^), the carbonyl content of each sample was estimated, and the results were expressed as nmol carbonyl/mg protein.

#### 2.4.6. Determination of Thiol Group

Ellman’s assay assessed free thiols in glycated samples with modest changes, as described by Ellman in 1959 [24]. First, 70 µL of the samples were incubated for 15 min with 5mM DTNB ( in 0.1 M PBS (pH 7.4) at 25 °C and absorbance of the samples was measured at 410 nm. From the L-cysteine (0–1.5 mM) standard, the concentration of free thiols was determined and expressed as nmol/mg protein.

#### 2.4.7. Total Fluorescent AGEs Formation Using Fluorescence Spectroscopy

The formation of fluorescent AGEs during albumin glycation is generally evaluated by monitoring their fluorescence at λex/λem of 370 nm and 440 nm, respectively, using a Agilent Cary Eclipse spectrofuorometer (Agilent Technologies, Victoria, Australia) [25]. The percent inhibition of glycation was calculated as follows: % Inhibition of glycation = Fluorescence of glycated BSA − Fluorescence of native orNSAE-treated BSA)/Fluorescence of glycated BSA × 100.

#### 2.4.8. Individual AGEs Formation

Some AGEs exhibit auto-fluorescence which can be detected at specific excitation and emission wavelengths [26]. Argpyrimidine, pentosidine, vesperlysine, and crossline were formed, which can be detected based on their characteristic fluorescence. Specifically, the samples’ concentration was 1.5 mg/mL (in sodium phosphate buffer 50Mm, pH 7.4). The emission spectra were collected across a wavelength range of 300–600 nm using a spectrofluorometer to estimate the amounts of argpyrimidine, pentosidine, vesperlysine, and crossline.

### 2.5. In Vitro Aggregation of BSA Induced by D-Ribose

#### 2.5.1. Congo Red Assay

The Congo red assay was used to examine the amyloid cross-structure. Glycated samples (50 µL) were mixed with 50 µL Congo red dye (100 µM) and incubated for 20 min. At 530 nm, the absorbance was measured [27].

#### 2.5.2. Thioflavin T Assay

Thioflavin T, an amyloid cross-structure marker, was quantified using a modified version of a previous approach [28]. To summarize, 100 µL of thioflavin T (64 µM) in 0.1 M PBS with a pH of 7.4 was added to the samples (10 µL) and then incubated for 30 min at 25 °C. Using a spectrofluorometer, the fluorescence intensity was measured at an excitation of 450 nm and an emission of 600 nm.

#### 2.5.3. Measurement of Amyloid Fibril Inhibition Using SDS-Polyacrylamide Gel Electrophoresis

To test the effect of NSAE on glycation-induced protein aggregation, standard sodium dodecyl sulfate-polyacrylamide gel electrophoresis (SDS-PAGE) was performed using the usual protocol [29]. Samples were placed onto gels with tracking dye (5% stacking and 10% resolving) and electrophoresed at 80 V. Gels were stained for 60 min with bromophenol blue dye, then destained using standard procedures before being imaged for analysis.

### 2.6. In Vitro Glycation of Plasmid DNA 

The ribose-mediated DNA damage was investigated using the previous approach [30]. The pBR322 (0.25 μg), Lysine (20 mM), D-ribose (250 mM), and FeCl3 (100 μM) were incubated in the presence/absence of NSAE in potassium phosphate buffer (100 mM; pH 7.4). The reaction mixtures were incubated at 37 °C for three hours. Samples were examined using 1% agarose gel electrophoresis and visualized in Gel-Doc after incubation. Using ImageJ, DNA pictures from Gel-Doc were analyzed to calculate the integrated density (IntDen). The dark bands and light backgrounds were created by inverting or converting the gel image’s colors. The software was used to generate a graph of the gray area against density. The IntDen was further calculated, and an increase/decrease in bands was deterined.

### 2.7. Statistical Analysis 

GraphPad Prism version 8.0 for Windows (GraphPad App, San Diego, CA, USA) was used for the analysis. The data were reported as Means ± SD, and the outcomes were derived from three independent experiments. The data were analyzed for significance using a two-way analysis of variance (ANOVA). Tukey’s multiple comparison tests were employed to ascertain whether treatments differed from one another, and the results are expressed as mean ± SD (*n* = 3). ^a^
*p* < 0.05 when compared to BSA, ^b^
*p* < 0.05 when compared to BSA+D-ribose.

## 3. Results

### 3.1. Plant Material Extraction

The percentage yield of the *N. sativa* seed extract obtained using the Soxhlet method was 16.56%.

### 3.2. Phytochemical Analysis

#### 3.2.1. Total Phenolic Content

Table 1 shows the total phenolic content of NSAE, and the values are derived from a calibration curve (y = 0.0033x ± 0.0289, R^2^ = 0.994) of gallic acid (5–500 µg/mL). The phenolic content in NSAE was 228.18 ± 0.013 GAE/g dry weight extract. Furthermore, the results of TPC content found in the present study were comparable to those reported by Thilakarathna and his co-workers in 2018 [31], who found that the TPC in the seeds was in the range of 89.53 to 437 g GAE/g of powdered seeds. 

#### 3.2.2. Total Flavonoid Content

Flavonoids are secondary antioxidant metabolites whose strength is determined by the amount and position of free-OH groups [32]. The total flavonoid content values were derived from the calibration curve (y = 0.0184 ± 0.0289, R^2^ = 0.9989) of quercetin (0–50 µg/mL), as shown in Table 1. The flavonoid content in NSAE was 191.644 ± 0.032 mg QE/g dry weight of the extract.

### 3.3. Enzyme Inhibition Assays

#### 3.3.1. Inhibition Assay for α-Amylase Activity (DNSA)

In vitro and in vivo amylase inhibitors reduce the activity of salivary and pancreatic amylase. When administered in excessive doses in the diet, they can affect animal growth and metabolism, although they may be helpful in treating obesity and diabetes [33]. Figure 1 presents the findings of the DNSA research. NSAE inhibits the enzyme at all doses, with a maximum value of 38.21% at a concentration of 1 mg/mL. The IC_50_ for the extract in the α-amylase enzyme inhibition assay was approximately 1.39 ± 0.016 mg/mL. The results of the α-amylase inhibition assay revealed that inhibitory activity was concentration-dependent. 

#### 3.3.2. Inhibition Assay for α-Glucosidase Activity

Maltase-glucoamylase and sucrase-isomaltase are two glucosidases. Each enzyme comprises two active subunits, one on the C-terminus and the other on the N-terminus of the original protein. All four subunits of α-glucosidases can hydrolyze maltose. Sucrase can only be hydrolyzed by the C-terminal component of sucrase-isomaltase [34]. The α-glucosidase inhibitory assay indicated that NSAE exhibited concentration-dependent inhibitory potential, as shown in Figure 1. We found an inhibitory activity of 47.33% in α-glucosidase at the maximum concentration of 1 mg/mL. The IC_50_ for the extract in the α-glucosidase enzyme inhibition assay was approximately 1.01 ± 0.022 mg/mL.

### 3.4. Analysis of Physicochemical Processes and Characterization

#### 3.4.1. Estimation of Hyperchromicity Using UV–Visible Spectroscopy

Our UV–Vis spectroscopic research results showed that D-ribose-glycated BSA’s structural changes create a rise in absorbance, which results in an increase in hyperchromicity as seen in the absorption spectra at 280 nm in comparison to the absorbance of native BSA (Figure 2). However, the addition of NSAE to the glycated sample demonstrated a significant drop in hyperchromicity, with a maximum decline of 13.38% found in the 0.1 mg/mL NSAE-treated sample compared to the glycated sample.

#### 3.4.2. Fourier Transform Infrared (FTIR) Spectrum

The FTIR spectrum analysis was accomplished to categorize the biomolecules present in the aqueous extract of *Nigella sativa* seeds (Figure 3). The results confirmed the presence of phenols and alcohols with a peak ratio at 3278.99 cm^−1^ corresponding to hydroxyl and O-H stretching frequency, respectively. The peak at 2171.85 cm^−1^ assigned to the C-H stretching indicates the presence of some alkane compounds; the peak at 1639.49 cm^−1^ confirms carboxylic acids; the peak value at 1516.05 cm^−1^ confirms nitro compounds; the peak value at 1342.46 cm^−1^ confirms alkanes; the peak value at 1266.30 cm^−1^ confirms aromatic amines; the peak value at 1230.58 cm^−1^ demonstrates alkyl halides; the peak value at 1018.41 cm^−1^ confirms aliphatic amines. Alkyl halides may also be seen in the peak at 964.41 cm^−1^.

#### 3.4.3. Measurement of Browning

Browning value increased as heating duration increased and was impressively high in BSA+D-ribose compared to BSA alone, as shown in Figure 4. The percentage inhibition of browning by NSAE at a 100 µg/mL concentration was 14.54%, 20%, 23.48%, and 27.34% in 7, 14, 21, and 28 days, respectively, for the glycated system. Furthermore, AG reduced the browning in the range of 23.66 to 37.41% in the D-ribose-induced glycation system, as mentioned in Table 2.

#### 3.4.4. Determination of Ketoamine Content

We discovered that NSAE lowered ketoamine production considerably. The results were expressed in 1-DMF concentration (1–8 mM) derived from a calibration curve (y = 0.0451x − 0.0323, R^2^ = 0.994), as shown in Figure 5. D-ribose-induced glycated protein had considerably greater ketoamine levels than BSA alone after each week of incubation. Ketoamine production was drastically reduced when NSAE (100 µg/mL) and AG (1 mM) were added. After a 28-day incubation period, NSAE reduced ketoamine production by 27.23%, whereas 1 mM AG inhibited ketoamine synthesis by 39.44% in the D-ribose-induced glycated system (Table 2).

#### 3.4.5. Determination of Protein Carbonyl Content

Throughout the investigation, the carbonyl content of the glycated sample was significantly higher than that of non-glycated samples. The results were expressed in terms of concentration (µM/mg protein) calculated using an extinction coefficient of 22,000, as shown in Figure 6. After 7, 14, 21, and 28 days of incubation, NSAE (100 µg/mL) reduced the protein carbonyl by 9.15%, 12.56%, 20.46%, and 31.86%, respectively. On the other hand, AG (1 mM) maximally decreased the carbonyl content of proteins in the range of 19.85% to 47.34% after 28 days of incubation (Table 2).

#### 3.4.6. Determination of Thiol Group

The glycation of proteins is linked to an increase in free radical generation. Conversely, free radicals can harm proteins by oxidizing the thiol group. Thus, thiol group oxidation in BSA is employed to demonstrate glycation-induced free radical production [35]. In this study, NSAE at a concentration of 100 µg/mL considerably reduced the depleting protein thiol group of glycated protein as shown in Figure 7. Compared to BSA+D-ribose at week one, the level of thiol groups at week four decreased by 38.69%. 

#### 3.4.7. Total Fluorescent AGEs Formation

The development of AGEs was observed by monitoring the fluorescence intensity of the BSA-D-ribose system every week. It was observed that the NSAE reduced the formation of AGEs because of Amadori products’ oxidative breakdown. Figure 8 illustrates that, as the incubation period increased, BSA-glycation with D-ribose increased considerably. After 28 days of incubation, the % suppression of AGE formation by NSAE at 100 µg/mL was 67.81% for the BSA-D-ribose system, while AG reduced the generation of AGEs by 77.32%.

#### 3.4.8. Individual AGE Formation

Pentosidine has a high absorption of 335 nm and a maximum emission of roughly 405 nm, whereas argpyrimidine absorbs at a wavelength of 320 nm and has a maximum emission of 380 nm [36]. As shown in Figure 9, at an excitation wavelength of 320, 335, 350, and 380 nm, BSA+ D-ribose increased fluorescence intensity more than BSA alone. In this study, the fluorescence spectra revealed that the quantities of argpyrimidine in the glycated sample decreased when NSAE was present. Furthermore, the concentration of fluorophores, i.e., vesperlysine and crossline—with excitation wavelengths of 335 and 380 nm with a fluorescence maximum of 385 and 440 nm, respectively—were also increased in the glycated system. After 28 days of incubation, it was found that the fluorescence intensity of each fluorescent AGE decreased in the presence of NSAE. 

### 3.5. In Vitro Aggregation of BSA Induced by D-Ribose

#### 3.5.1. Congo Red Assay

Compared to glycated BSA, NSAE significantly reduced secondary structural modifications in BSA, as shown in Figure 10A. After 28 days of incubation, the maximum significant decrease in amyloid cross-structure formation was observed in the presence of NSAE and AG, i.e., 40.06% and 52.97%, respectively. We demonstrated that NSAE could prevent aggregation by preventing the transition from α-helix to cross-β conformer, suggesting that they may play a suppressive role in the late stage (post-Amadori) of BSA glycation.

#### 3.5.2. Thioflavin T Assay

To see if NSAE acts as an inhibitor of the amyloid-like aggregates, we added a fluorescent reagent, i.e., ThT. In the presence of BSA incubated with D-ribose, ThT fluorescence increased considerably. Compared to AG, NSAE (100 µg/mL) therapy had a significant anti-aggregation potential. Compared to non-glycated BSA, glycated BSA demonstrated a four-fold increase in amyloid cross-conformation (Figure 10B). Furthermore, NSAE reduced the degree of formation of amyloid cross-structure in a time-dependent way. After 28 days of incubation, the maximum significant decline in the formation of amyloid cross-structure in BSA+D-ribose was observed in NSAE and AG, i.e., 51.44% and 63.58%, respectively.

#### 3.5.3. Measurement of Amyloid Inhibition Using SDS-Polyacrylamide Gel Electrophoresis

After prolonged glycation, protein may form micelle-like aggregates. Glycation of albumin caused physicochemical changes in protein conformation, molecular weight, and pH, all of which were in the direction of transitioning from an α-helical to a β-sheet structure and the production of nano-fibrillar amyloids [37]. In the glycated system (Lane 2), the SDS gel pattern confirmed protein build-up and enhanced crosslinking. Increased components revealed the relative abundance of high molecular weight protein chains. As shown in Figure 11, NSAE (Lane 6) and AG (Lane 4) prevented D-ribose-induced aggregation formation and changes in its characteristics. In both, the structural alterations indicated a reduction in cross-linking

### 3.6. In Vitro Glycation of Plasmid DNA

The pBR322, plasmid DNA, is a sensitive sensor of single-strand breaks caused by glycation and free radical damage. Glycation of DNA leads to the creation of Amadori products, which produce specific nucleotide adducts indicative of a variety of aberrant situations, such as oxidative stress [38]. DNA-AGEs may contribute to the loss of genomic integrity as people face age-related problems. Figure 12 shows the effects of NSAE on DNA damage mediated by the glycation of D-ribose with lysine in the presence of Fe^+3^. The pBR322 alone displayed two bands, band I indicates the linear form, and band II indicates the supercoiled form (Figure 12B, C). Adding D-ribose and lysine to pBR322 increased the breakage of DNA strands by increasing the intensity of the open circular form of bands. However, the addition of lysine, D-ribose, FeCl_3_, and NSAE/AG to pBR322 did not result in DNA cleavage, i.e., pBR322 remained in its supercoiled form. These results were further analyzed using ImageJ software (Version 1.53s. https://imagej.nih.gov/ij/ (accessed on 9 September 2022)), and it can be observed that 31.73% of DNA gets converted from supercoiled form to linear shape in the ribose-induced glycation system (Table 3).

## 4. Discussion

Several studies have found that accumulating cross-linked AGEs on long-lived proteins may have a role in developing diabetes and age-related problems. Furthermore, the severity of these issues is reflected in serum advanced glycation end-products, whereas treatment strategies aim to block or delay the progression of AGE formation. *N. sativa* has shown many biological properties, including antidiabetic properties [39]. According to our findings, *Nigella sativa* aqueous extract (NSAE) inhibits the α-amylase and α-glucosidase enzymes at all doses, with the maximum value of 38.21% and 47.33%, respectively, seen at a concentration of 1000 µg/mL. These results agreed with the results of Tiji et al. (2021) [40], who reported that the acetone extract (75.8 ± 0.36% inhibition) had higher α-amylase inhibition than hexane extract (58.0 ± 10.86% inhibition) and quite like the standard acarbose (at 5 µM). A previous study by Sobhi et al. (2016) [41] showed that the % inhibition of α-glucosidase by two polar lipid fractions of *N. sativa* was reflected in their IC_50_ (0.51 ± 0.04 mg/mL and 0.55 ± 0.09 mg/mL, respectively) when compared to thymoquinone (0.65 ± 0.05 mg/mL) and acarbose (0.53 ± 0.06 mg/mL). FTIR spectroscopic analysis of NSAE was given in Figure 2, revealing the presence of different functional groups of bioactive compounds in the form of peaks. The FTIR spectral analysis revealed that the NSAE possesses a wide range of bioactive compounds such as alcohols, phenols, primary and secondary amines, carboxylic acids, nitro compounds, etc., and these compounds are involved in the biological activities of NSAE.

We demonstrated that NSAE substantially prevented ribose-induced fluorescent AGEs production in a time-dependent manner. In addition, NSAE lowered ketoamine levels, which was linked to reduced development of AGEs, showing that it has an inhibitory effect on Amadori production and conversion to AGEs. Our studies are supported by a previous study that showed that absorbance increased substantially on the first day before reaching a plateau. After the reaction of D-ribose with BSA, the increase in absorbance indicates the production of advanced glycation end products [42]. The fluorescence spectral analysis revealed that the quantities of argpyrimidine and pentosidine in the glycated sample decreased when NSAE was present. 

Protein oxidation was detected in the current study by increasing the carbonyl concentration of protein and depleting the thiol group of protein. NSAE, on the other hand, influenced the reduction in carbonyl content concentration and thiol group oxidation in the D-ribose-induced glycation system. After 28 days of incubation, NSAE reduced the amount of protein carbonyl by 31.86% and improved thiol groups by 38.59%. In a similar study by Rubab and her colleagues [9], it was found that the carbonyl concentration of the glycated sample was reduced by 39.79% in methanolic extract and 30.79% in aqueous extract of *N. sativa* seeds. Compared to other phenolic-rich extracts, our data imply that *N. sativa* has antiglycation properties. 

Protein glycation, in general, directly impacts the production of protein aggregation. Insoluble aggregates can create an amyloid cross-structure, causing protein structure and stability to be altered. The present findings show that NSAE inhibits the production of amyloid cross-structures in BSA. This favorable effect may aid in lowering the risk of diabetes complications. 

The phytochemical analysis revealed that NSAE is high in phenolic and flavonoid compounds. Previous illustrations demonstrated that phenolic-rich plant extracts prevented sugar-induced protein glycation [43]. During the glycation process, bioactive constituents like ferulic acid [44,45], rutin [46], gallic acid [47], carvacrol [48], salicylic acid [49], and kaempferol [50] may exert their glycation inhibition actions by neutralizing reactive carbonyl intermediates, scavenging free radicals, and chelating redox-inducing transition metal ions [51]. The antiglycation activity of NSAE on SDS-PAGE and the reduction in AGE production points to a feasible scenario in which the phenolics prevented both early and later phases of ribose-induced BSA glycation.

Glycation causes partial unwinding and/or fragmentation of the double helix, according to more extensive investigations of DNA stability and dynamics [52]. Damaged plasmid DNA is depicted as an open circular band, while the major band corresponds to a supercoiled form of DNA. Our findings show that residues of transition metals that produce free radicals may be responsible for ribose-mediated DNA breakage. The duration of Fe3+ treatment and the presence of ferric ions exacerbated the damage to plasmid DNA. In biological systems, trace metals such as iron may react with H_2_O_2_ to form hydroxyl radicals, which then cause DNA strand breaks [38].

Numerous toxicological research on *N. sativa* have been conducted, and it has been observed that NSAE does not show any toxicity at lower doses. No significant changes in liver function were observed when hepatic enzyme levels and histological alterations in liver tissue were evaluated after oral administration of an aqueous extract of *N. sativa* seeds [53]. *N. sativa* did not have any harmful effects on liver enzymes in asthmatic patients, according to Al Ammen and his colleagues [54]. According to another study, *N. sativa* doses up to 1.0 g/kg body weight did not produce liver toxicity or create hepatic damage or obstructive hepatobiliary disease [55].

According to our findings, glycation-mediated secondary complications of diabetes, protein aggregation-mediated neurological diseases, and glycoxidative DNA damage can all be prevented with *Nigella sativa* aqueous extract. According to the literature, the findings showed a substantial correlation between the plant’s antioxidative activities and its antiglycation capabilities. In a study by Burits and Bucar (2000) [56], the essential oil from these seeds was extracted and then analyzed using TLC for its antioxidant activity; t-Anethole, thymoquinone, terpineol, and carvacrol were discovered to have strong radical scavenging activities. The medicinal benefits of *N. sativa* oil and extracts have already been described. The inhibitory potential of *N. sativa* at various stages of glycation was discovered by Dalli et al., 2022 [57]. They showed significant suppression of the early and late stages of AGEs production. Thymoquinone may reduce the absorption of the main carbonyl compounds identified in AGEs obtained from food, according to Losso et al. (2011) [58]. *N. sativa* reduces the synthesis of ketoamine implicated in the formation of AGE by preventing glucose autoxidation [59]. *N. sativa* inhibits AGE, although the exact mechanism by which it does so is yet unknown. This mechanism needs to be determined by a more thorough investigation. In addition, further research is required to determine the actual mechanism of NSAE inhibition and the active chemicals involved in the process.

## 5. Conclusions

The current research explains how *Nigella sativa* seed extract prevents albumin glycation, oxidative changes, and amyloid aggregation induced by D-ribose. D-ribose takes part in the glycation of proteins, creating AGEs that cause structural and functional changes in biomolecules like proteins and nucleic acids. This study revealed that NSAE protects against D-ribose-mediated glycation in vitro. NSAE inhibits α-amylase and α-glucosidase, suggesting that it could be used as an antidiabetic drug, particularly in postprandial hyperglycemic circumstances. NSAE also decreases ketoamine levels, AGE production, and amyloid formation. In BSA, it also lowers the carbonyl concentration of the protein and prevents the thiol group from being modified. The positive effects of NSAE could be used to avoid or control AGE-mediated diseases, especially in people with diabetes. More study in animal models is required to understand the antiglycation properties of NSAE fully.

## Figures and Tables

**Figure 1 medicina-58-01816-f001:**
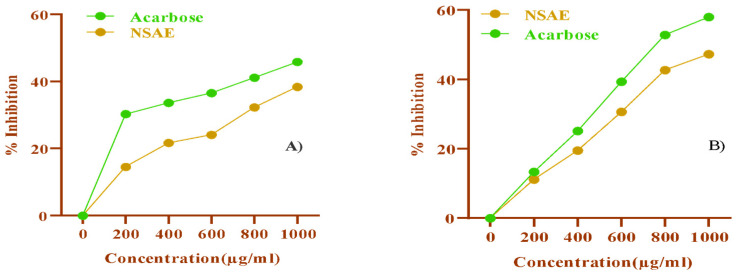
(**A**) α-amylase percent inhibition and (**B**) α-glucosidase percent inhibition in the presence of NSAE, utilizing acarbose as a standard (0–1000 µg/mL). The IC_50_ for NSAE in the α-amylase and α-glucosidase enzyme inhibition assays were approximately 1.39 ± 0.016 mg/mL and 1.01 ± 0.022 mg/mL, respectively. Results are expressed as the mean ± SD (*n* = 5).

**Figure 2 medicina-58-01816-f002:**
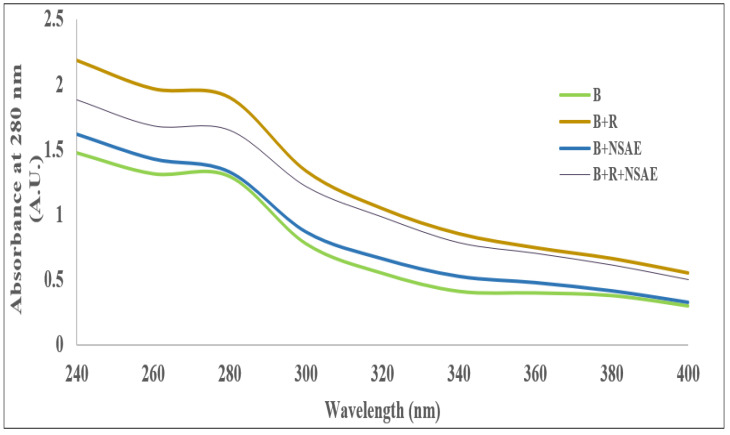
Effect of *N. sativa* seed extract on D-ribose glycation-induced change in the hyperchromicity pattern. The decrease in absorption was observed in the presence of NSAE.

**Figure 3 medicina-58-01816-f003:**
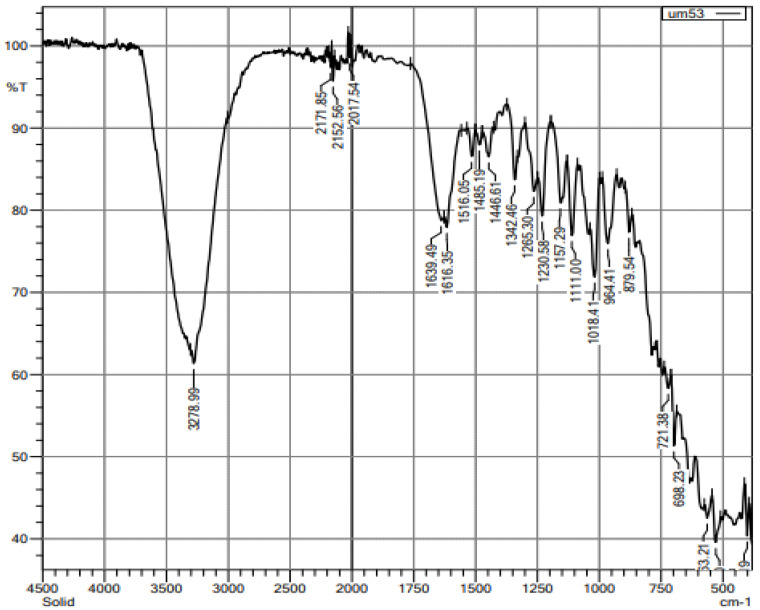
FTIR spectra of the aqueous extract of *N. sativa* seeds. The X-axis represents the wavenumber (cm^−1^), and Y-axis represents the % transmittance.

**Figure 4 medicina-58-01816-f004:**
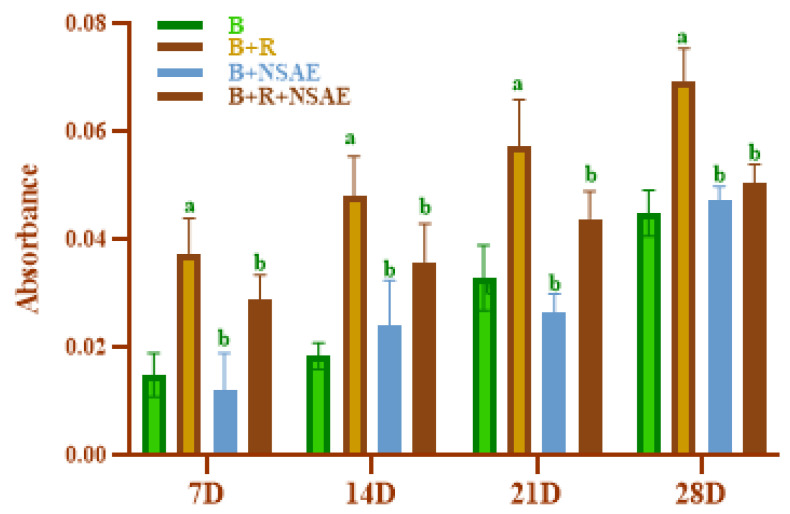
The effect of NSAE on browning reaction in BSA+D-ribose system. B-BSA, B+R=BSA+Ribose, B+NSAE=BSA+*Nigella sativa* aqueous extract, and B+R+NSAE= BSA+Ribose+*Nigella sativa* aqueous extract. Results are represented as mean ± SD (*n* = 5). ^a^
*p* < 0.05 when compared to BSA and ^b^
*p* < 0.05 when compared to BSA+D-ribose.

**Figure 5 medicina-58-01816-f005:**
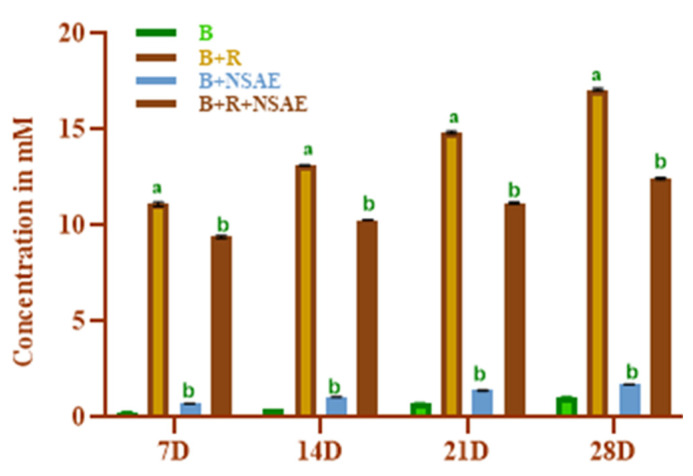
The effect of NSAE on the formation of ketoamine in the BSA+D-ribose system. B-BSA, B+R=BSA+Ribose, B+NSAE=BSA+*Nigella sativa* aqueous extract, and B+R+NSAE= BSA+Ribose+*Nigella sativa* aqueous extract. Results are represented as mean ± SD (*n* = 5). ^a^
*p* < 0.05 when compared to BSA and ^b^
*p* < 0.05 when compared to BSA+D-ribose.

**Figure 6 medicina-58-01816-f006:**
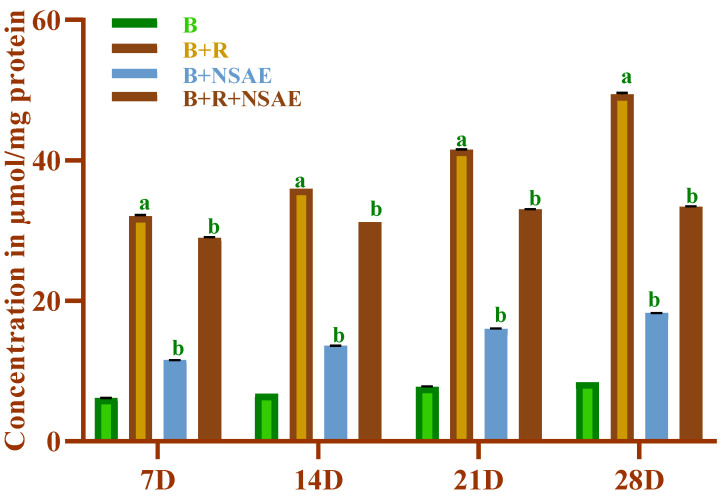
The effect of NSAE on carbonyl group generation in BSA+D-ribose system. B-BSA, B+R=BSA+Ribose, B+NSAE=BSA+*Nigella sativa* aqueous extract, and B+R+NSAE= BSA+Ribose+*Nigella sativa* aqueous extract. Results are represented as mean ± SD (*n* = 5). ^a^
*p* < 0.05 when compared to BSA and ^b^
*p* < 0.05 when compared to BSA+D-ribose.

**Figure 7 medicina-58-01816-f007:**
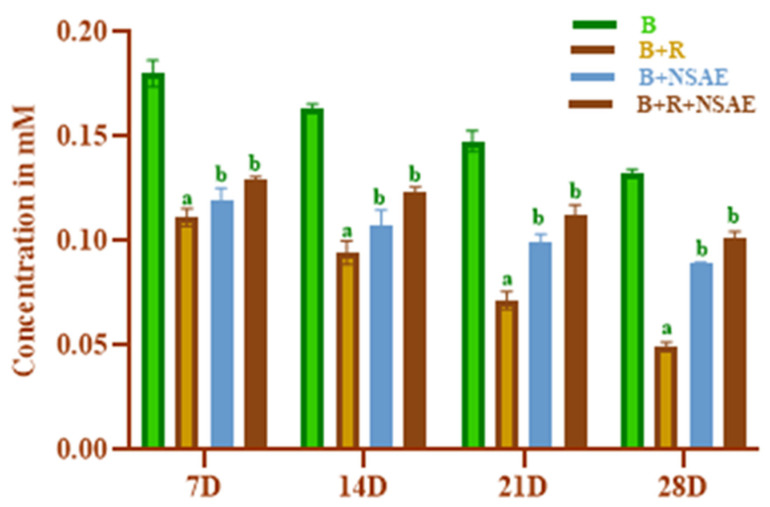
The effect of NSAE on thiol group formation in BSA+D-ribose system. B-BSA, B+R=BSA+Ribose, B+NSAE=BSA+*Nigella sativa* aqueous extract, and B+R+NSAE= BSA+Ribose+*Nigella sativa* aqueous extract. Results are represented as mean ± SD (*n* = 5). ^a^
*p* < 0.05 when compared to BSA and ^b^
*p* < 0.05 when compared to BSA+D-ribose.

**Figure 8 medicina-58-01816-f008:**
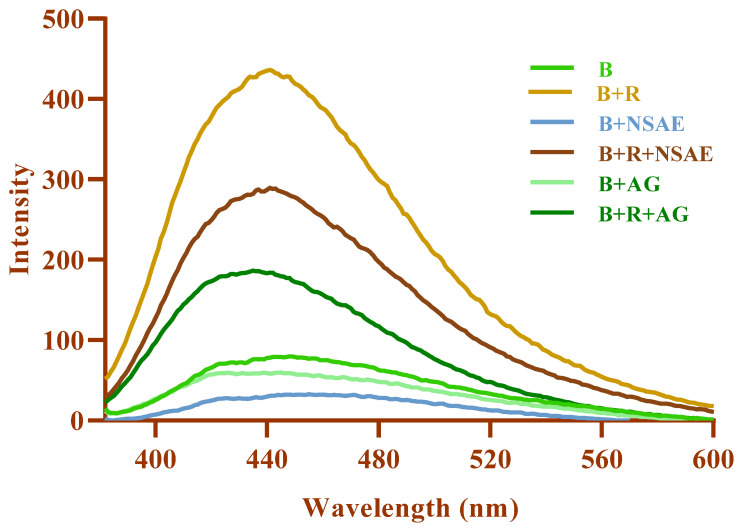
The effects of *Nigella sativa* aqueous extract (NSAE) on fluorescent AGEs formation in BSA+D-ribose system.

**Figure 9 medicina-58-01816-f009:**
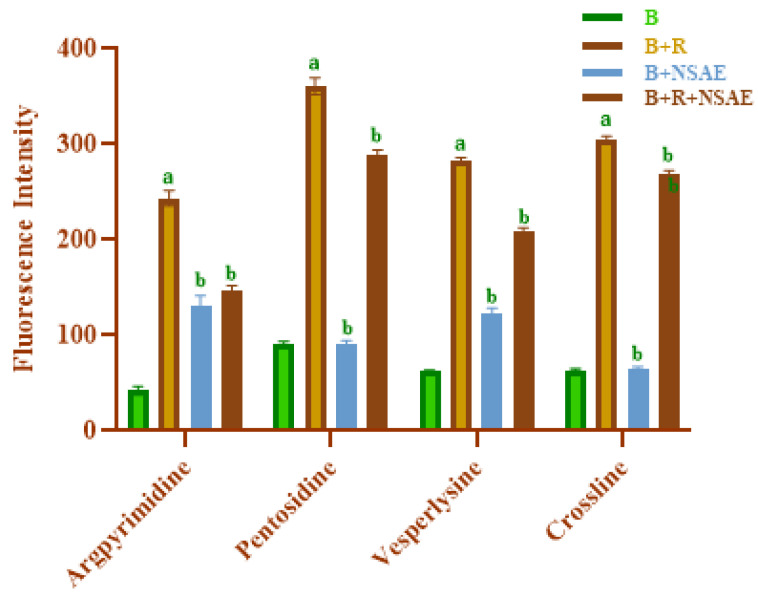
*Nigella sativa* aqueous extract (NSAE) affects individual AGEs formation in the BSA+D-ribose system. Results are represented as mean ± SD (*n* = 5). ^a^
*p* < 0.05 when compared to BSA and ^b^
*p* < 0.05 when compared to BSA+D-ribose.

**Figure 10 medicina-58-01816-f010:**
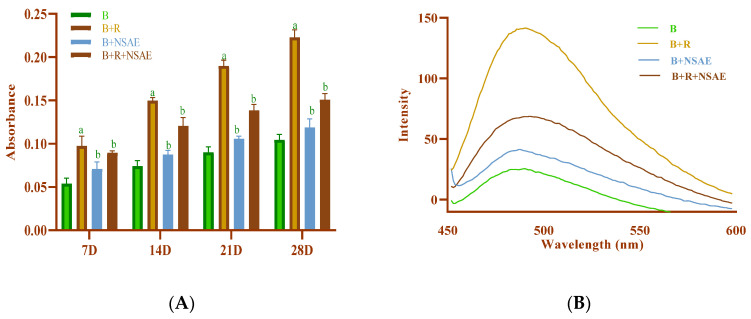
The effect of NSAE on the formation of amyloid cross β-structure using (**A**) Congo red assay and (**B**) Thioflavin T assay in BSA+D-ribose system. Results are represented as mean ± SD (*n* = 5). ^a^
*p* < 0.05 when compared to BSA and ^b^
*p* < 0.05 when compared to BSA+D-ribose.

**Figure 11 medicina-58-01816-f011:**
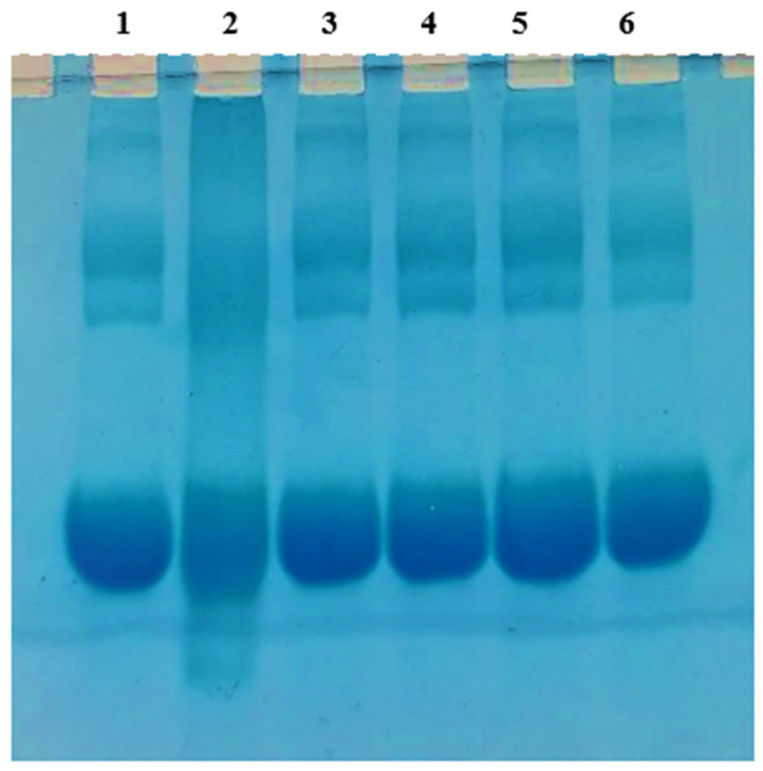
SDS-PAGE of the effects of NSAE on protein aggregation is mediated by glycation of BSA with D-ribose after 28 days of incubation. Samples were incubated with the following: Lane 1: BSA alone; Lane 2: BSA+D-ribose; Lane 3: BSA+AG; Lane 4: BSA+ D-ribose+ AG; Lane 5: BSA+NSAE; Lane 6: BSA+ D-ribose+ NSAE.

**Figure 12 medicina-58-01816-f012:**
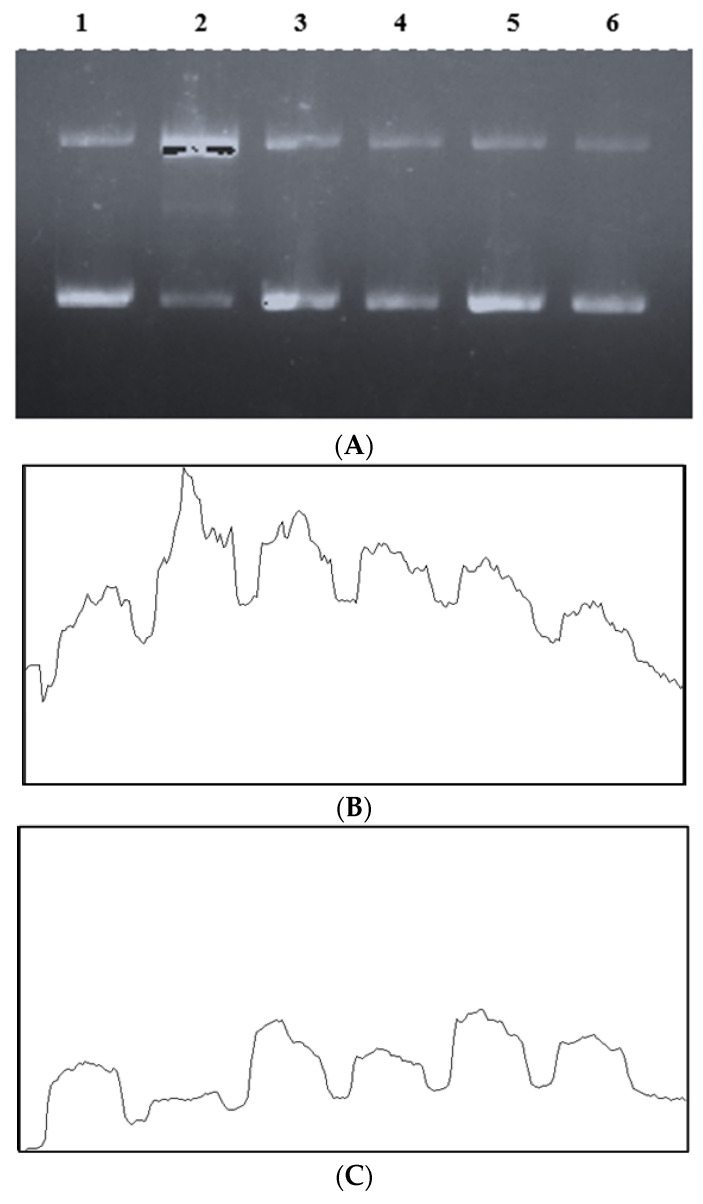
Agarose gel electrophoresis of plasmid DNA. (**A**) The effects of NSAE on DNA damage induced by glycation of D-ribose with lysine in the presence of Fe+2. Samples were incubated as follows: Lane 1: DNA; Lane 2: DNA+Lysine+D-ribose+FeCl_3_ (GS); Lane 3; DNA+AG; Lane 4: DNA+GS+AG; Lane 5: DNA+NSAE; Lane 6: DNA+GS+NSAE. The X-axis in (**B**) and (**C**) represents the density, and Y-axis represents the gray value.

**Table 1 medicina-58-01816-t001:** Total phenolic and flavonoid content of *Nigella sativa* aqueous extract (NSAE). Data represented as mean ± SD of triplicate tests.

Test Sample	TPC (mg/GAE g DW)	TFC (mg QE/g DW)
*Nigella sativa*	228.18 ± 0.013	191.644 ± 0.031

**Table 2 medicina-58-01816-t002:** The percentage inhibition of AG and NSAE on protein glycation and aggregation in the BSA+D-ribose system.

% Inhibition	Browning	Ketoamine Content	Protein Carbonyl Content	Thiol Group	Congo Red Assay
AG	37.41%,	39.44%	47.34%	47.90%	52.97%
NSAE	27.34%	27.23%	31.86%	38.69%	40.06%

**Table 3 medicina-58-01816-t003:** Percent increase or decrease in the integrated density of electrophoretic bands.

Band I			Band II		
Lane	Integrated Density	Percent Increase/Decrease	Lane	Integrated Density	Percent Increase/Decrease
**1**	149,351	0	**1**	119,767	0
**2**	160,126	+7.21	**2**	81,765	−31.73
**3**	134,268	+10.10	**3**	91,443	−23.65
**4**	144,438	+3.29	**4**	85,715	−28.43
**5**	128,280	+14.11	**5**	93,658	−21.80
**6**	144,551	+3.21	**6**	111,581	−6.83

The increase is represented with a (+) sign, while the decrease is represented with a (−) sign.

## Data Availability

Not applicable.

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
