# Peer review of "D-Ribose-Induced Glycation and Its Attenuation by the Aqueous Extract of Nigella sativa Seeds"

_medicina, 2022, doi:10.3390/medicina58121816_

Round 1

Reviewer 1 Report

The manuscript describes the potential antidiabetic effect of the aqueous extract of Nigella sativa (NSAE) seeds. Through in vitro experiments the authors showed that NSAE reduces the activity of α-amylase and α-glucosidase enzymes. Besides, anti-glycation activity of NSAE was also observed in assays of D-ribose-induced BSA glycation. Finally, the anti-glycation activity was associated to the reduction of protein aggregation, an event that occurs after the protein glycation.

In a quick search for papers on Nigella sativa, it was possible to observe that studies about the effects of aqueous extracts of seeds on D-ribose-induced glycation are scarce. Thus, the present manuscript could give a special contribution for evidences of the biological activities of aqueous extract of N. sativa seeds with special focus on diabetes.

Below the authors can find some considerations about different aspects of the manuscript.

1. The authors cite several works describing different effects of Nigella sativa (bronchodilator, diuretic, immune-modulator). However, it is not clear if those effects come from medicinal use of N. sativa. In Introduction section, authors should include more information about N. sativa such as its medicinal/ popular use as well as its distribution among different regions around the world.

2. In the Abstract, the authors reported the inhibitory concentrations 50% (IC50) of the extract in the enzyme assays (α-amylase and α-glucosidade; line 20-21). However, this information was not found in the description of the results (Results section), neither in the methodological description. Authors must describe the analysis performed to determine the IC50 and this information must be described in the results. As suggestion, IC50 value could be included in legend of both graphs in Figure 1.

3. In browning, ketoamine content, protein carbonyl content, thiol group, and Congo red assays, the authors used aminoguanidine (AG) as positive control. The results of AG were presented separately in Table 2. To enrich results presentation, the authors should include the results of NSAE in Table 2. This can allow better comparison between the effects of NSAE and the reference compound (AG).

Why did the authors decide to present only the results of 28 days of AG on Table 2?

4. In browning, ketoamine content, protein carbonyl content, thiol group, and Congo red assays, the authors used 100 μg/mL (line 266, 308) of NSAE. In general, dose-response curve experiments are performed to find an ideal concentration. Why did the authors decide to use the specific concentration of 100 μg/mL?

5. In the figure 6, in the 7D (B+R+NSAE) group, inside the column there is a green letter “b”. Please, remove it.

6. In the figure 11, please adjust the numbers to their respective lanes on figure.

7. In Table 3, adjust column width to accommodate the word “Lane” in one line. In the same table, zero (0) should be centralized in column.

8. In “Discussion” section the authors described that the higher effect of NSAE was observed at 100 μg/mL (line 410-412). However, in the figure 1 they observed that the higher effect was obtained of 1000 μg/mL of NSAE. Please, explain this inconsistence.

9. The authors performed FTIR analysis to elucidate possible compounds present in NSAE. However, there was not discussion about the contribution of this experiment to elucidation of bioactive compounds present in NSAE. The authors should include a discussion about the contribution of FTIR experiments in elucidation of phytochemical that could be contributing to observed inhibitory activity on α-amylase and α-glycosidase and anti-glycation effects.

10. We can find a variety of articles reporting the anti-diabetic effects of plant-derived bioactive compounds. In this regard, how can NSAE be advantageous over other extract of plants? To enrich the Discussion, authors should include a short description about the possible advantages of using NSAE.

11. Nigella sativa should be written in italic. Please, replace Nigella sativa for Nigella sativa or N. sativa

Author Response

                                       Response to Reviewer 1 Comments

We are very much thankful to the editor and anonymous reviewers for their depth comments, suggestions, and corrections, which have greatly improved the manuscript. I hope the suggested revision of the manuscript has significantly improved the level of satisfaction. All the reviewers' comments have been taken into account, and the manuscript has been revised. The corrections are highlighted in yellow text.

Reviewer(s)' Comments to Author:

Point 1: The authors cite several works describing different effects of Nigella sativa (bronchodilator, diuretic, immune-modulator). However, it is unclear if those effects come from the medicinal use of N. sativa. In the Introduction section, authors should include more information about N. sativa, such as its medicinal/ popular use as well as its distribution among different regions around the world.

Response 1: As‌ ‌per‌ ‌the‌ ‌suggestions‌ ‌given,‌ the introduction section ‌of‌ ‌the‌ ‌manuscript‌ ‌has‌ ‌been‌ edited. ‌ 

  1. sativa is native to the eastern Mediterranean, the Indian subcontinent, northern Africa, and Southwest Asia [10]. In addition to being well known for its culinary uses, it has a long history of use in traditional medicine. In traditional medicine, N. sativa has been recommended for a variety of illnesses and disorders like hypertension, anorexia, amenorrhea, paralysis, dermatitis, and bronchitis [11]. These traditional applications of N. sativa are largely attributed to its various medicinal benefits like antidiabetic, anti-inflammatory, antihypertensive, antioxidant, antimicrobial, immunomodulatory, cardioprotective, anticancer, neuroprotective, nephroprotective, hepatoprotective, and gastroprotective properties [12,13]. The black cumin plant is considered one of the most promising options to be utilized in preventing and treating diabetes because of its potent ability to lower blood sugar by raising insulin levels [14].

Point 2: In the Abstract, the authors reported 50% (IC50) inhibitory concentrations of the extract in the enzyme assays (α-amylase and α-glucosidase; lines 20-21). However, this information was not found in the description of the results (Results section), nor in the methodological description. Authors must describe the analysis performed to determine the IC50 and this information must be described in the results. As a suggestion, the IC50 value could be included in the legend of both graphs in Figure 1.

Response 2: As‌ ‌per‌ ‌the‌ ‌suggestions‌ ‌given,‌ the inhibitory concentrations of 50% (IC50) of the extract in the enzyme assays (α-amylase and α-glucosidase) ‌has‌ ‌been‌ reported in the results section.

The IC50 for the extract in the α-amylase enzyme inhibition assay was approximately 1.39±0.016 mg/ml. The IC50 for the extract in the α-glucosidase enzyme inhibition assay was approximately 1.01±0.022 mg/ml.

 It has been mentioned in the methodology section that GraphPad Prism (version 8) was used for the analysis of IC50 values. The data were analyzed for significance using a two-way analysis of variance (ANOVA).

Point 3: In browning, ketoamine content, protein carbonyl content, thiol group, and Congo red assays, the authors used aminoguanidine (AG) as a positive control. The results of AG were presented separately in Table 2. To enrich the presentation of the result, the authors should include the results of NSAE in Table 2. This can allow a better comparison between the effects of NSAE and the reference compound (AG). Why did the authors decide to present only the results of 28 days of AG in Table 2?

Response 3: As per the suggestion given, the results of NSAE have been included in Table 2. Glycation is a slow process and takes time to form advanced glycation end products (AGEs). In most of the reported literature, an incubation period of four weeks has been used to analyze the formation of glycation products and the inhibitory effect of any compound/extracts. Various reports suggested that the maximum inhibition was found on the 28th day of incubation. Our study also got the maximum inhibition (37.41%) after four weeks. That is why only the results of 28 days of AG have been presented in Table 2.

% Inhibition

Browning

Ketoamine content

Protein carbonyl content

Thiol group

Congo red

assay

AG

37.41%,

39.44%

47.34%

47.90%

52.97%

NSAE

27.34%

27.23%

31.86%

38.69%

40.06%

Point 4: In browning, ketoamine content, protein carbonyl content, thiol group, and Congo red assays, the authors used 100 μg/mL (line 266, 308) of NSAE. In general, dose-response curve experiments are performed to find an ideal concentration. Why did the authors decide to use the specific concentration of 100 μg/mL?

Response 4: We decided to use the specific 100 μg/mL concentration based on dose-response curve experiments. All these glycation assays (browning, ketoamine content, protein carbonyl content, thiol group, and Congo red assays) were performed using different concentrations of NSAE i.e., from 50 μg/mL to 500 μg/mL and it was seen that the maximum inhibition was reported at a concentration of 100 μg/mL. For example, total fluorescent AGE production inhibition was 59.42%, 67.81%, and 63.94% at a concentration of 50 μg/mL, 100 μg/mL, and 500 μg/ml respectively.

Point 5: In figure 6, in the 7D (B+R+NSAE) group, inside the column, there is a green letter “b”. Please, remove it.

Response 5: The green letter has been removed from figure 6.

Point 6:  In figure 11, please adjust the numbers to their respective lanes on the figure.

Response 6: The numbers to their respective lanes in figure 11 have been adjusted.

Point 7: In Table 3, adjust the column width to accommodate the word “Lane” in one line. In the same table, zero (0) should be centralized in the column.

Response 7: In Table 3, the column width to accommodate the word “Lane” in one line and zero (0) in the column has been adjusted.

Point 8: In the “Discussion” section the authors described that the higher effect of NSAE was observed at 100 μg/mL (lines 410-412). However, in figure 1 they observed that a higher effect was obtained at 1000 μg/mL of NSAE. Please, explain this inconsistency.

Response 8: The higher effect of NSAE in the enzyme assays (α-amylase and α-glucosidase) was observed at 1000 μg/mL and the antiglycation effects were observed at 100 μg/mL. In the “Discussion” section the higher effect of NSAE observed at 100 μg/mL has been written mistakenly. The mistake has been rectified.

Point 9: The authors performed FTIR analysis to elucidate possible compounds present in NSAE. However, there was no discussion about the contribution of this experiment to the elucidation of bioactive compounds present in NSAE. The authors should include a discussion about the contribution of FTIR experiments in the elucidation of phytochemicals that could be contributing to observed inhibitory activity on α-amylase and α-glycosidase and anti-glycation effects.

Response 9: Glycation is a non-enzymatic reaction between the free amino (-NH2) group in proteins and the carbonyl group of reducing sugars, leading to the generation of Amadori products. So, the FTIR technique has been used to identify different functional groups.

From the FTIR analysis, it was revealed that the NSAE possesses a wide range of bioactive compounds such as alcohols, phenols, primary and secondary amines, carboxylic acids, nitro compounds, etc. that correspond to various significant phytochemicals. These bioactive components from NSAE are highly beneficial as a curative agent for several diseases ranging from cancer to acquired immune deficiency syndrome.

Point 10: We can find a variety of articles reporting the anti-diabetic effects of plant-derived bioactive compounds. In this regard, how can NSAE be advantageous over another extract of plants? To enrich the Discussion, the authors should include a short description of the possible advantages of using NSAE.

Response 10: As per our previous research, the aqueous extract of N. sativa seeds exhibited the highest values: TPC, TFC, DPPH, ABTS, hydroxyl radical scavenging activity, and nitric oxide scavenging activity as compared to the other seed extracts of N. sativa (methanolic, ethanolic and hexane extract. The aqueous fraction of N. sativa seeds are high in antioxidative and anti-inflammatory bioactive components, allowing it to be developed as a diabetes nutraceutical. Furthermore, the presence of extracts had a substantial antioxidant effect, protecting DNA against the oxidative stress agent, H2O2.

Prairna Balyan and Ahmad Ali (2022). Comparative analysis of the biological activities of different Nigella sativa L. seeds extracts. Ann. Phytomed., 11(1):577-587. http://dx.doi.org/10.54085/ap.2022.11.1.67.

The discussion section has been revised accordingly and the changes incorporated are : Nigella sativa has been subjected to several toxicological studies, and it has been reported that at a lower dose, NSAE does not report any toxicity. The oral administration of an aqueous extract of Nigella sativa seeds showed no significant changes in liver function, evaluating hepatic enzymes level as well as histopathological changes of liver tissue (Mohammed, 2010). Al Ammen and his colleagues (2003) reported no toxic effects of Nigella sativa on hepatic enzymes among asthmatic patients. Another study by Dollah et al., 2013 indicated that a Nigella sativa dose of up to 1.0 g/kg body weight showed no hepatocellular damage or obstructive hepatobiliary disease and did not cause toxicity to the liver.

Point 11: Nigella sativa should be written in italic. Please, replace Nigella sativa with Nigella sativa or N. sativa

Response 11: As per the suggestion, the modification has been done.

Reviewer 2 Report

Reviewer suggestions

-Introduction doesn’t provide information about the traditional use of the plant.

-All scientific names of plants species need to be in italic Nigella sativa.

 - in vitro, in vivo terms need to be in italic.

-Nigella sativa or commonly known as the miracle herb or black cumin has been studies extensively. Several works have reported the antidiabetic and antiglycation properties of the NS seeds Please do not hessite to cite them.

-As mentioned in your article, the seeds were cleaned using water which a highly polar molecule that could cause a significant lost of polar bioactive compounds present in NS seeds.

Did you confirm that the washing process doesn’t affect the chemical composition of the plant?

-I totally agree that using soxhlet apparatus that is based on reflux extraction technique is very adequate in order to extract the majority of the molecules. Also, the yield of extraction will much higher. However, in your case you used water in soxhlet apparatus that in order to be refluxed it needs to be evaporated at a temperature of 100°C. Thus, the high temperature could be able to have a negative effect on the chemical composition of the extract by causing a degradation of NS bioactive compounds. 

- Please precise time for the extraction.

-Normally, the yield is calculated by dividing the weight of the extract by initial plant’s weight used in extraction the outcome multiplied by 100.

In the equation you did mention the theoretical yield, how did you calculate that theoretical yield.

-Concerning the methods section: ALL protocols need to be detailed explicitly in order to facilitate their reproducibility. Also, the concentrations used in each test for the aqueous extract and the standards.

-Line 27: check the author name in reference 27.

-Check the units in table 1 for TPC mgGAE/ g DW.

-Table 1 needs to be placed after section 3.2.2.

- Some studies have reported that the aqueous extract is rich with polyphenolic compounds (Phenolic acid+ flavonoids) while the flavonoids were found in a very small portion which was contradictory to your findings where the extract is rich with important amount of flavonoids how could you explain that.

- Section 3.2.2: we noticed that the highest concentration was 1mg/mL for a-amylase that showed an inhibition percentage above 40%. Some evidence showed that at a concentration of about 1.82mg/mL an inhibition percentage of about 82% was recorded.

-Section 3.4: We also noticed that the authors decided to use FTIR technique as a characterization technique for the aqueous extract, but the reviewer see that this technique doesn’t report much information about the chemical composition of NS aqueous extract. It’s very recommended to report HPLC analysis data which gives more accurate results on the chemical composition.

-It will be very helpful for the reader to report the IC50 values for a-amylase and a-glycosidase.

-Put Table 2 after section 3.5.1.

-Line 295: please insert the different concentrations tested.

-Line 297: At what concentration the aminoguanidine (AG) induced a decrease of carbonyl content.

-Figure 11: Please align the figure with the numbers above.

-Line 410: Nigella sativa was studies exhaustively, thus several studies have reported the utility of aqueous extract as antidiabetic agent because of its ability to decrease glycemia or by inhibiting digestive enzymes or by inhibition intestinal glucose absorption. We notice that none of these studies were mentioned on your manuscript.

-IC50 à IC50  (50 as a subscript number)

-Line 418: In your manuscript you reported the effect of thymoquinone that is a terpen compound found mainly in Nigella sativa essential oil and hexane extract. Does any study report the presence of Thymoquinone in aqueous extract? If yes please cite that paper.

-Line 445: During the glycation process, bioactive 445 constituents may exert their glycation inhibition actions …..

Please name these bioactive compounds.

-Line 448: Delete electrophoresis to avoid repetition.

-Line 456: Fe3+ or Fe2+.

-H2O2 à H2O2

- Line 461: Suggest a mechanism of action for the obtained activity.

-Compare the obtained results with the results found in other studies.

-Add a paragraph where you demonstrate that the aqueous extract is nontoxic or less toxic??

Please check these articles. They could be helpful for you to improve your manuscript.

 https://doi.org/10.1155/2021/9979419

https://doi.org/10.3389/fphar.2022.1036129

Author Response

                                  Response to Reviewer 2 Comments

We are very much thankful to the editor and anonymous reviewers for their depth comments, suggestions, and corrections, which have greatly improved the manuscript. I hope the suggested revision of the manuscript has significantly improved the level of satisfaction. All the reviewers' comments have been taken into account, and the manuscript has been revised. The corrections are highlighted in yellow text.

Point 1: Introduction doesn’t provide information about the traditional use of the plant.

Response 1: The introduction section ‌of‌ ‌the‌ ‌manuscript‌ ‌has‌ ‌been‌ edited.

The changes are incorporated as follows:

  1. sativa is native to the eastern Mediterranean, the Indian subcontinent, northern Africa, and Southwest Asia [10]. In addition to being well known for its culinary uses, it has a long history of use in traditional medicine. In traditional medicine, N. sativa has been recommended for various illnesses and disorders like hypertension, anorexia, amenorrhea, paralysis, dermatitis, and bronchitis [11]. These traditional applications of N. sativa are largely attributed to its various medicinal benefits like antidiabetic, anti-inflammatory, antihypertensive, antioxidant, antimicrobial, immunomodulatory, cardioprotective, anticancer, neuroprotective, nephroprotective, hepatoprotective, and gastroprotective properties [12,13]. The black cumin plant is one of the most promising options that can be utilized in the prevention and treatment of diabetes because of its potent ability to lower blood sugar by raising insulin levels [14].

Point 2: All scientific names of plant species need to be in italics Nigella sativa, - in vitro, and in vivo terms need to be in italic.

Response 2: The terms have been italicized.

Point 3: Nigella sativa or commonly known as the miracle herb or black cumin has been studied extensively. Several works have reported the antidiabetic and antiglycation properties of the NS seeds. Please do not hessite to cite them.

Response 3: As per the suggestion given, the following references have been added to the manuscript.

Dalli M, Daoudi NE, Abrigach F, Azizi S-e, Bnouham M, Kim B and Gseyra N (2022) In vitro α-amylase and hemoglobin glycation inhibitory potential of Nigella sativa essential oil, and molecular docking studies of its principal components. Front. Pharmacol. 13:1036129. doi: 10.3389/fphar.2022.1036129.

Balyan, P.; Khan, J.; Ali, A. Therapeutic Potential of Nigella Sativa in the Prevention of Aggregation and Glycation of Proteins. In Black Seeds (Nigella sativa): Pharmacological and Therapeutic Applications; 2021. https://doi.org/10.1016/B978-0-12-824462-3.00015-9.

 Toma, C. C.; Olah, N. K.; Vlase L.; Mogosan, C.; Mocan, A. Comparative studies on polyphenolic composition, antioxidant, and diuretic effects of Nigella sativa L. (Black cumin) and Nigella damascene L. (Lady-in-a-mist) seeds. Molecules 2015, 20(6). doi:10.3390/molecules20069560. –

Al-Ghamdi, M. S. Protective effect of Nigella sativa seeds against carbon tetrachloride-induced liver damage. American Journal of  Chinese Medicine. 2003;31(5):721–8.\

Balyan P.; Ali, A. Comparative analysis of the biological activities of different extracts of Nigella sativa L. seeds. Annals of  Phytomedicine 11(1):577-587. http://dx.doi.org/10.54085/ap.2022.11.1.67.

Point 4: As mentioned in your article, the seeds were cleaned using water which is a highly polar molecule that could cause a significant loss of polar bioactive compounds present in NS seeds. Did you confirm that the washing process doesn’t affect the chemical composition of the plant?

Response 4: Extraction is a vital first step in the study of medicinal plants since it is necessary to extract the appropriate chemical components from plant materials for subsequent examination. The results of the present study are similar to the data reported by Khattak et al. (2008), in which the extraction yields of water extract were comparable to our extract yield. Compared to our findings, according to Dalli et al. (2021), the aqueous extract has the most polyphenols (51.63 ± 1.95) mg GAE/mg DW. Compared to acarbose, the aqueous extract had a significantly good a-amylase inhibitory action compared to other extracts. The correlation between antioxidant assays and total phenolic content was investigated, and the antioxidant assays revealed a significant positive relationship. These findings indicate that there was not any significant loss of bioactive compounds in NS seeds during extraction.

Prairna Balyan and Ahmad Ali (2022). Comparative analysis of the biological activities of different extracts of Nigella sativa L. seeds. Ann. Phytomed., 11(1):577-587. http://dx.doi.org/10.54085/ap.2022.11.1.67.

Point 5: I totally agree that using the Soxhlet apparatus that is based on the reflux extraction technique is very adequate to extract most of the molecules. Also, the yield of extraction will be much higher. However, in your case, you used water in the Soxhlet apparatus that in order to be refluxed needs to be evaporated at a temperature of 100°C. Thus, the high temperature could be able to have a negative effect on the chemical composition of the extract by causing the degradation of NS bioactive compounds. 

- Please precise time for the extraction.

Response 5: The temperature used for the evaporation was 50-60°C, so there wasn’t any potential loss in the chemical composition of the extract. The time for extraction using the Soxhlet apparatus was 24 hours.

Point 6:  In figure 11, please adjust the numbers to their respective lanes on the figure.

Response 6: The numbers to their respective lanes in figure 11 have been adjusted.

Point 7: Normally, the yield is calculated by dividing the weight of the extract by the initial plant’s weight used in extraction the outcome multiplied by 100. In the equation you did mention the theoretical yield, how did you calculate that theoretical yield?

Response 7:  The amount of product predicted by stoichiometry is called the theoretical yield, whereas the amount obtained is called the actual yield.

The equation we used for calculating the percent yield is as follows:

Percent yield=(Dry weight of extract ÷ Dry weight of plant material) × 100. We have revised the equation.

Point 8: Concerning the methods section: All protocols need to be detailed explicitly to facilitate their reproducibility. Also, the concentrations used in each test for the aqueous extract and the standards.

Response 8: Methods have been revised, and concentrations have been added as per the suggestion.

Point 9: Check the author’s name in reference 27.

Response 9: The author’s name has been checked and rectified.

Point 10: Check the units in table 1 for TPC mg GAE/ g DW.

Response 10: The revision has been made.

Point 11: Table 1 needs to be placed after section 3.2.2.

Response 11: As per the suggestion, the modification has been done.

Point 12: Some studies have reported that the aqueous extract is rich with polyphenolic compounds (Phenolic acid+ flavonoids) while the flavonoids were found in a very small portion which was contradictory to your findings where the extract is rich with an important amount of flavonoids how could you explain that.

Response 12: Results of chemical analysis showed that flavonoid contents varied between the tested spices' extracts obtained using the same extraction solvent. The variations in TFC results between the current and prior research could be attributable to differences in the extraction solvent, plant component employed, analysis method, environmental stress, and climatic and geographical variables. Most probably, the variation in phytochemical contents of these spices is attributed to their difference in genetic makeup. According to a study shown by Sasikumar et al., (2020) total phenolics content was equivalent to flavonoids, in the extracts of N. sativa.

Sasikumar JM, Erba O, Egigu MC. In vitro antioxidant activity and polyphenolic content of commonly used spices from Ethiopia. Heliyon. 2020 Sep 22;6(9):e05027. doi: 10.1016/j.heliyon.2020.e05027. PMID: 32995654; PMCID: PMC7511827.

Point 13: Section 3.2.2: we noticed that the highest concentration was 1 mg/mL for a-amylase, which showed an inhibition percentage above 40%. Some evidence showed that at a concentration of about 1.82mg/mL an inhibition percentage of about 82% was recorded.

Response 13: The researchers used Nigella sativa essential oil in the evidence showing an inhibition percentage of about 82%, at a concentration of about 1.82mg/ml. Compared to our findings, there is also a piece of evidence that reported that the aqueous extract of N. sativa seeds was able to inhibit the α-amylase in a dose-dependent manner, and an inhibition capacity of 84% at a concentration of 100 mg/ml. It is showing that the extract is showing inhibition at a very high dose. This indicates that the biological activities of the plant extracts vary from time to time and region to region depending upon the polarity, extraction solvent, extraction method, and various other factors.

Sathiavelu, A.; Sangeetha, S.; Archit, R.; Mythili, S. In Vitro Anti-Diabetic Activity of Aqueous Extract of the Medicinal Plants Nigella Sativa, Eugenia Jambolana, Andrographis Paniculata and Gymnema Sylvestre. International Journal of Drug Development and Research 2013, 5 (2).

Point 14: Section 3.4: We also noticed that the authors decided to use FTIR technique as a characterization technique for the aqueous extract, but the reviewer sees that this technique doesn’t report much information about the chemical composition of NS aqueous extract. It’s very recommended to report HPLC analysis data, which gives more accurate results on the chemical composition.

Response 14: A number of reports are there which indicated the HPLC analysis of N. sativa seed extracts and a lesser number of analyses using FTIR. Glycation is a non-enzymatic reaction between the free amino (-NH2) group in proteins and the carbonyl group of reducing sugars, leading to the generation of Amadori products. So, the FTIR technique has been used to identify different functional groups.

From the FTIR analysis, it was revealed that the NSAE possesses a wide range of bioactive compounds such as alcohols, phenols, primary and secondary amines, carboxylic acids, nitro compounds, etc. that correspond to various significant phytochemicals. These bioactive components from NSAE are highly beneficial as a curative agent for several diseases ranging from cancer to acquired immune deficiency syndrome.

Point 15: It will be very helpful for the reader to report the IC50 values for a-amylase and a-glycosidase.

Response 15: IC50 values for a-amylase and a-glycosidase have been reported in the results section.

The IC50 for the extract in the α-amylase enzyme inhibition assay was approximately 1.39±0.016 mg/ml. The IC50 for the extract in the α-glucosidase enzyme inhibition assay was approximately 1.01±0.022 mg/ml.

Point 16: Put Table 2 after section 3.5.1.

Response 16: The modification has been done.

Point 17: Line 295: Please insert the different concentrations tested.

Response 17: The concentrations have been inserted at all the relevant places.

Point 18: Line 297: At what concentration the aminoguanidine (AG) induced a decrease in carbonyl content?

Response 18: Aminoguanidine (AG) at a concentration of 1 mM has been used throughout the glycation process. 

Point 19: Figure 11: Please align the figure with the numbers above.

Response 19: The revision has been done.

Point 20: Line 410: Nigella sativa was studied exhaustively, thus several studies have reported the utility of aqueous extract as an antidiabetic agent because of its ability to decrease glycemia by inhibiting digestive enzymes or by inhibiting intestinal glucose absorption. We notice that none of these studies were mentioned in your manuscript.

Response 20: As per the suggestion given, the references have been added.

Dalli, M; Daoudi, N. E.; Azizi, S.; Benouda, H.; Bnouham, M.; Gseyra, N. "Chemical composition analysis using HPLC-UV/GC-MS and Inhibitory Activity of Different Nigella sativa fractions on pancreatic α-amylase and intestinal glucose absorption", BioMed Research International, 2021, Article ID 9979419.

Aazza, S.; El-Guendouz, S.; Miguel M. G. Antioxidant, anti-inflammatory and anti-hyperglycaemic activities of essential oils from thymbra capitata, thymus albicans, thymus caespititius, thymus carnosus, thymus lotocephalus and thymus mastichina from Portugal,” Natural Product Communications, 2016,11(7),1029–1038.

 El Rabey, H.A., Al-Seeni, M.N.,  Bakhashwain, A.S.  The Antidiabetic Activity of Nigella sativa and Propolis on Streptozotocin-Induced Diabetes and Diabetic Nephropathy in Male Rats. Evidence-based complementary and alternative medicine. 2017, eCAM, 2017, 5439645. https://doi.org/10.1155/2017/5439645.

Arasteh, A.; Farahi, S.; Habibi-Rezaei, M.; Moosavi-Movahedi, A. A. Glycated Albumin: An Overview of the In Vitro Models of an In Vivo Potential Disease Marker. Journal of Diabetes and Metabolic Disorders. 2014. https://doi.org/10.1186/2251-6581-13-49.

Point 21: IC50 à IC50  (50 as a subscript number)

Response 21: Thank you. It has been corrected.

Point 22: Line 418: In your manuscript, you reported the effect of thymoquinone which is a terpene compound found mainly in Nigella sativa essential oil and hexane extract. Does any study report the presence of Thymoquinone in the aqueous extract? If yes, please cite that paper.

Response 22: Yes, there are studies that have reported the presence of thymoquinone in the aqueous extract of N. sativa seeds. However, the content of thymoquinone is less in the aqueous extract as compared to the methanolic extract. The references have been added to the manuscript.

Mohamad, N.S.; Muhamud, N.A.S.A.; Amran, S.I. Screening of Thymoquinone (Tq) Content in Nigella sativa-Based Herbal Medical Products. Journal of Natural Product and Plant Resources, 2018, 8 (3): 41-45.

Point 23: Line 445: During the glycation process, bioactive constituents may exert their glycation inhibition actions. Please name these bioactive compounds.

Response 23: NSAE contains various bioactive constituents, and these constituents are involved in the process of glycation as mentioned in the discussion section with references.

During the glycation process, bioactive constituents like ferulic acid  (Narasimhan and M. Chinnaiyan, 2009; Zheng, et al., 2020), rutin (Dubey et al., 2017), gallic acid (Adefegha et al., 2015), carvacrol (Aazza et al., 2016), salicylic acid (Dalli, et al., 2021) and Kaempferol Toma et al., 2015) may exert their glycation inhibition actions by neutralizing reactive carbonyl intermediates, scavenging free radicals, and chelating redox-inducing transition metal ions.

Point 24: Line 448: Delete electrophoresis to avoid repetition.

Response 24: Deleted

Point 25: Line 456: Fe3+ or Fe2+, H2O2 à H2O2

Response 25:  It should be Fe+3 and H2O2. The corrections have been done.

Point 26: Line 461: Suggest a mechanism of action for the obtained activity. Compare the obtained results with the results found in other studies. Add a paragraph where you demonstrate that the aqueous extract is nontoxic or less toxic.

Response 26: The discussion part has been modified as per the suggestion.

According to the literature, the results indicated that the antiglycation properties significantly correspond to the antioxidative activities of the plant. In a study by Burits & Bucar (2000), the essential oil was extracted from these seeds and then tested for their antioxidant activity using TLC. And found that t-anethole, thymoquinone, terpineol, and carvacrol showed significant radical scavenging properties. It has been already mentioned that N. sativa oil as well as its extracts have wide therapeutic properties. Dalli and his colleagues (2022) revealed the inhibitory potential of N. sativa at different stages of glycation.  They demonstrated strong inhibition of the early and late stages of AGEs formation. According to Losso et al. (2011), thymoquinone may decrease the absorption of chief carbonyl compounds found in AGEs derived from food. N. sativa prevents the autoxidation of glucose, inhibiting the formation of ketoamines implicated in AGE formation (Mahmood et al., 2013). The mechanism involved in AGE inhibition by N. sativa is still unclear and needs more comprehensive research to identify the mechanism.

Losso, J.N., Bawadi, H.A., Chintalapati, M., 2011. Inhibition of the formation of advanced glycation end products by thymoquinone. Food Chem. 128 (1), 55.

Mahmood, T., Moin, S., Faizy, A.F., Naseem, S., Aman, S., 2013. Nigella Sativa as an antiglycating agent for Human Serum Albumin. Int. J. Sci. Res. 2 (4), 25–27.

Burits, M. and Bucar, F. (2000), Antioxidant activity of Nigella sativa essential oil. Phytother. Res., 14: 323-328. https://doi.org/10.1002/1099-1573(200008)14:5<323::AID-PTR621>3.0.CO;2-Q

Dalli M, Daoudi NE, Abrigach F, Azizi S-e, Bnouham M, Kim B and Gseyra N (2022) In vitro α-amylase and hemoglobin glycation inhibitory potential of Nigella sativa essential oil, and molecular docking studies of its principal components. Front. Pharmacol. 13:1036129. doi: 10.3389/fphar.2022.1036129.

Nigella sativa has been subjected to several toxicological studies, and it has been reported that NSAE doesn’t report any toxicity at a lower dose. The oral administration of an aqueous extract of Nigella sativa seeds showed no significant changes in liver function evaluating hepatic enzymes level as well as histopathological changes of liver tissue (Mohammed, 2010). Al Ammen and his colleagues reported no toxic effects of Nigella sativa on hepatic enzymes among asthmatic patients. Another study by Dollah et al., 2013 indicated that a Nigella sativa dose of up to 1.0 g/kg body weight showed no hepatocellular damage or hepatobiliary obstructive disease and did not cause toxicity to the liver.

Round 2

Reviewer 2 Report

I would like to thank the authors for their commitment to making the modifications necessary in order to improve their MS. I encourage the publication of this article in the present work.